# A Time Mode Pulse Interleaver in a Silicon-on-Insulator Platform for Optical Analog-to-Digital Converters

**DOI:** 10.3390/mi15030326

**Published:** 2024-02-27

**Authors:** Donghe Tu, Zezheng Li, Yuxiang Yin, Huan Guan, Zhiguo Yu, Lifei Tian, Lei Jiang, Yuntao Li, Zhiyong Li, Zhongchao Fan

**Affiliations:** 1Institute of Semiconductors, Chinese Academy of Sciences, Beijing 100083, China; 2Key Laboratory of Optoelectronic Materials and Devices, Chinese Academy of Sciences, Beijing 100083, China; 3College of Materials Science and Opto-Electronic Technology, University of Chinese Academy of Sciences, Beijing 100049, China; 4School of Integrated Circuits, University of Chinese Academy of Sciences, Beijing 100049, China

**Keywords:** silicon photonics, pulse interleaver, time mode interleaving

## Abstract

We propose and demonstrate a novel on-chip optical sampling pulse interleaver based on time mode interleaving. The designed pulse interleaver was fabricated on a 220 nm silicon-on-insulator (SOI) platform, utilizing only one S-shaped delay waveguide. Interleaving is achieved by the relative time delay between different optical modes in the waveguide, eliminating the need for any active tuning. The total length of the delay waveguide is 5620.5 µm, which is reduced by a factor of 46.3% compared with previously reported time-wavelength interleaver schemes. The experimental results indicate that the device can convert an optical pulse into a 40 GHz pulse sequence composed of four pulses with a root mean square (RMS) timing error of 0.9 ps, making it well suited for generating high-frequency sampling pulses for optical analog-to-digital converters.

## 1. Introduction

Analog-to-digital converters (ADCs), which perform the data conversion of analog signals into digital signals, play a key role in signal processing applications such as ultrawideband communication [1], wireless communications [2], software radio [3] and radar systems [4]. The development trend of electronic ADCs is gradually slowing down as the improvement in electronic components guided by Moore’s law has become slower over the last decade, facing electrical bottlenecks such as aperture jitter and clock Jitter. Therefore, photonic ADCs are a strong candidate for high-speed analog-to-digital conversion [5,6,7,8], owing to the inherent characteristic of the high speed and low latency of photons.

For the photonic sampled ADCs, an optical sampling signal with a high frequency is a key factor to achieving high-speed sampling in the system. To achieve high-frequency optical signals, various methods have been proposed. Microwave photonic-based electro-optical modulation can output high-frequency optical signals [9,10,11]. However, the pulse width is relatively large, which is not suitable for the sampling source. To address the issue, a mode-locked laser or an electro-optic comb system [12,13] can be used as the sub-picosecond width pulse source. Nevertheless, it is still challenging to directly produce high repetition rate optical pulses. The output repetition rate of the majority of commercial mode-locked lasers is from tens of MHz to several GHz. Electro-optic comb systems require microwave sources of the same high frequency. To generate optical pulses of tens of GHz, an ultra-fast low-jitter microwave source is required, which is not cost-efficient. Therefore, an additional optical multiplexing block after the mode-locked laser or an electro-optic comb system to increase the pulse repetition rate is essential.

The optical time interleaver (TI) scheme utilizes asymmetric Mach–Zender interferometers to split, delay, and combine the optical signals [14,15]. It is a simple and effective solution for multiplexing the optical pulses. But each combining stage induces 3 dB of optical loss. To overcome this issue, a scheme based on a time-wavelength interleaver (TWI) [16,17,18,19] was proposed, and it is widely used to generate high-frequency optical sampling pulses. In the TWI scheme, the original pulse signal is first separated into independent waveguide channels by a demultiplexer. The wavelength signals in every channel propagate in the independent waveguide to achieve a relative time delay. Then, the multi-wavelength pulse signals are combined by a wavelength multiplexer to form a high-frequency pulse train [18]. In the TWI scheme, one can see that the number for the delay waveguide equals the number of wavelength channels, which makes it difficult to keep the device in a small area. It is noted that the phenomenon of wavelength shift exists in wavelength multiplexers and demultiplexers, such as in micro-ring resonators and cascaded Mach–Zehnder interferometers. Therefore, some active devices may be needed to tune the channel wavelengths, causing additional power consumption.

In this work, a novel scheme to generate a high-frequency sampling pulse train based on multiple modes in waveguides is proposed and demonstrated experimentally. Our approach employs a time mode interleaver (TMI) designed to establish relative time delays between four different modes in the silicon-on-insulator (SOI) waveguide. The device is composed of a power splitter, asymmetric directional couplers, and a delay waveguide. Compared with the reported TWI scheme, there is only one main delay path in our scheme based on our proposed S-shaped delay waveguide design. This design allows multiple modes to propagate in a single waveguide, effectively reducing the delay waveguide’s length by 46.3%. The proposed TMI comprises passive components, eliminating the need for additional power consumption. The experimental results demonstrate the separation of one optical pulse into four independent pulse signals with time delays of 0 ps, 24.4 ps, 47.7 ps, and 74.6 ps. This indicates that the frequency of the input pulse train with 10 GHz can be increased to 40 GHz after the manipulation of time mode interleaving. The root mean square (RMS) timing error is calculated to be 0.9 ps. The proposed TMI has the potential for generating high-frequency sampling pulses for optical analog-to-digital converters.

## 2. Design of the TMI

Figure 1 shows the block diagram of the optical sampling pulse interleaver using a TMI based on four TE modes. The original pulse train from a mode-locked laser is first fed into a 1 × 4 power splitter, and the four output signals propagate into the delay waveguide section at different locations. Then, one signal directly propagates to the start port (point A in Figure 1) of the delay waveguide and has the maximum transmission time. The other three output signals of the 1 × 4 power splitter are coupled into the delay waveguide at different locations (points B, C, and D in Figure 1) by mode couplers and propagate through different higher-order modes. Here, we chose asymmetric directional couplers to achieve mode transformation from the output of the power splitter to the delay waveguide, which is indicated by the dotted boxes in Figure 1. The schematic drawing of the asymmetric directional coupler is plotted in Figure 2.

In this work, we select a strip waveguide with 2 µm cladding on a 220 nm SOI platform. A cross-sectional view of the waveguide can be seen in Figure 3a. It is important to note that in a multi-mode waveguide, different optical modes exhibit distinct group indexes. Consequently, the group velocities of these different modes vary, allowing the formation of time delays as the multiple modes propagate through the waveguide.

We used a waveguide with a cross-section of 1900 nm × 220 nm as an example. Figure 3b shows the calculated group indexes of the four TE modes. At 1550 nm, the group indexes for the TE0, TE1, TE2, and TE3 modes were 3.7575, 3.8735, 4.0895, and 4.44885, respectively. From 1500 nm to 1600 nm, the group indexes of the higher modes were larger than those of the lower modes. The group index of the TE0 mode had the minimum value. Therefore, the corresponding pulse signal had the fastest propagation velocity according to the following equation: (1)$$t=\frac{L}{v_{g}}=\frac{L\times n_{g}}{c}$$
where $n_{g}$ and $v_{g}$ are the group index and group velocity, respectively, *L* and *t* are the transmission length and transmission time in the waveguide, respectively, and *c* is the speed of light in a vacuum. For a specified mode in the waveguide, a desired transmission time can be achieved by designing the corresponding propagation length. Notably, the presence of multiple modes in the same multi-mode waveguide allowed us to achieve the desired delay with just one delay waveguide in our proposed scheme.

The optical delay waveguide was constructed using several waveguides with different widths. The design includes multiple sections, each with a specific width. The width of the first section of the delay waveguide ($wg_{1}$ in Figure 1) was 450 nm, which was the width of the single-mode waveguides in the device. The path of the first delay waveguide had an S-shape, which offers a specified length for mode propagation. At the end of the first delay waveguide, a taper was used to increase the waveguide width from 450 nm to 930 nm, which was the width of the second delay waveguide ($wg_{2}$ in Figure 1). At location B in Figure 1, the pulse signal transmits through the first asymmetric directional coupler and completes the mode conversion from TE0 to TE1. Tapers were used at the end of each subsequent section of the delay waveguide, achieving the transition to wider waveguides in the next section. It is important to note that Euler waveguide bends were incorporated to minimize bending crosstalk and insertion loss. Three asymmetric directional couplers are required in the TMI design shown in locations B, C, and D in Figure 1. The widths of the input waveguides ($W_{\mathrm{in}}$ shown in Figure 2) of the three asymmetric directional couplers were set to be 450 nm, and the corresponding effective index of the TE0 mode was 2.352. When the effective index of a higher-order mode in the upper waveguide (shown in Figure 2) equals the effective index of the TE0 mode in the lower waveguide, the phase-matching condition is satisfied, and efficient coupling between the modes can be achieved. The effective indexes of the modes in the strip waveguides with different widths can be calculated [20]. The widths of $W_{\mathrm{out}}$ were chosen to be 900 nm, 1416 nm, and 1900 nm, making the effective indexes of the TE1, TE2, and TE3 modes, respectively, close to 2.352. The gap was set to be 200 nm. The lengths $L_{c}$ can be calculated using coupled mode theory [21], and they were optimized to be 28.5 μm, 36.0 μm, and 43.0 μm, respectively, by a commercial finite-difference time domain solver (FDTD, Ansys). The simulated conversion efficiencies of the three asymmetric directional couplers were 95.7%, 96.9%, and 98.6% at 1550 nm, corresponding to insertion losses of 0.19 dB, 0.14 dB, and 0.07 dB. The proposed asymmetric directional couplers could operate well when small wavelength deviations occurred. The simulation results show the three designed couplers exhibited ≤0.3 dB of loss across the C band. To further achieve a broadband and fabrication-tolerant mode conversion, adiabatic directional couplers with a tapered waveguide design [22] could be used.

In our design of the TMI, there are four TE modes when the pulse signals propagate to the end of the delay waveguide (point E in Figure 1). The reaching time was different for each mode, resulting in the frequency of the output pulse train of the TMI being four times the original pulse frequency from the input laser. The four lowest-order modes in TE polarization were chosen to minimize the required waveguide width. A wider waveguide has the risk of causing higher mode excitations, increasing mode crosstalk, and requiring a larger bending radius, which makes the layout more difficult. Here, the original pulse frequency was set to be 10 GHz with a period of 100 ps. If the relative time delay is fixed between different modes, then the time delay should be 0 ps, 25 ps, 50 ps, and 75 ps for the signals of the TE3, TE2, TE1, and TE0 modes, respectively, to obtain a 40 GHz output. The operating wavelength was 1550 nm in our design. The widths of the higher-order mode delay waveguides were designed to follow the above-mentioned $W_{\mathrm{out}}$ required by the asymmetric directional couplers and were set to be 930 nm, 1416 nm, and 1900 nm. The optical modes in the delay waveguide would propagate through waveguides with different widths. For example, the TE0 mode would transmit through $wg_{1}$ (450 nm width), $wg_{2}$ (930 nm width), $wg_{3}$ (1416 nm width), and $wg_{4}$ (1900 nm width). Therefore, the group indexes of the multiple modes in the four waveguides at 1550 nm should be calculated.

The symbol of × in Table 1 indicates that the modes were not supported in the corresponding waveguide with the specific width. It is clear that the relative time delay of the TE3 mode had a minimum value of zero in our design. Thus, the length of the waveguide with a 1900 nm width could be set to zero. The propagation time of the TE2 mode should be 25 ps, which was decided by the length of the waveguide with a width of 1416 nm. According to Equation (Equation 1), the calculated length of $wg_{1416\mathrm{nm}}$ is 1692.2 µm. Then, the propagation time in $wg_{1416\mathrm{nm}}$ of the TE0 and TE1 modes could be calculated, and so on. Table 2 shows the distribution of the time delay of different modes in different sections of the waveguide. Therefore, the lengths of $wg_{930\mathrm{nm}}$ and $wg_{450\mathrm{nm}}$ were set to be 1869.9 µm and 2058.4 µm, respectively. It is worth mentioning that a wavelength shift would affect the group indexes of the modes as shown in Figure 3. The slight group index variation within ±3% across the C band may result in small deviations in the delay time when the wavelength changes.

The total length of the S-shaped delay waveguide added up to be 5620.5 µm. For a TWI scheme with the same incrementing of the signal repetition rate from 10 GHz to 40 GHz, three single-mode delay waveguides are required, and the lengths should be 1743.1 µm, 3486.2 µm, and 5229.3 µm, which add up to 10,458.6 µm. Therefore, our proposed TMI scheme with an S-shaped waveguide decreased the total delay length by a factor of 46.3%.

## 3. Fabrication and Experimental Results

The proposed TMI was fabricated on a 220 nm SOI platform with a buried layer thickness of 3 µm. The pattern of the optical structure was defined using electron beam lithography (EBL) with a ZEP520A photoresist (Zeonrex Electronic Chemicals Company Tokyo, Japan). Then, the silicon layer was fully etched by inductively coupled plasma (ICP) etching. Figure 4 shows the microscope view and scanning electron microscope (SEM) view of the fabricated device. To simplify the testing process, there was only one output grating coupler in the testing device. At the end of the delay waveguide (point E in Figure 1), a mode demultiplexer was used to extract the TE0, TE1, and TE2 modes from the output multi-mode waveguide and convert them to TE0 modes, which were combined by three optical combiners. Because the length of $wg_{3}$ (1900 nm width) was zero, we eliminated one asymmetric directional coupler as a proof of concept. The corresponding output signal of the 1 × 4 MMI power splitter was directly connected to the optical power combiner.

Figure 5 shows the experimental set-up and the view of the testing environment. Due to lacking a suitable mode-locked laser, we used a bit pattern generator (BPG, SHF 12103A, Berlin, Germany) and a commercial modulator to generate an optical pulse with a wavelength of 1550 nm. The generated optical pulse is shown in Figure 6a, and the y axis represents the optical power. The distortion of the optical pulse was mainly caused by the limited electro-optical bandwidth of the BPG and the link, as well as the nonlinearity of the large signal response of the modulator.

The input optical pulse signal was coupled into the on-chip device by a grating coupler. After the propagation in the TMI, the output pulse signals were coupled out from the device and detected by an oscilloscope (Tek DSA8300, Beaverton, U.S.). Polarization controllers were used to control the light state before the modulator and the chip device. The experimental results of the TMI are presented in Figure 6b.

In the testing results of the pulse generator, the input optical pulse signal was divided into four pulses which were separated in the time domain. By examining the peak of each pulse signal, the relative time delays of the four optical pulses were measured to be 0 ps, 24.4 ps, 47.7 ps, and 74.6 ps, with a calculated RMS time error of 0.9 ps. This indicates that the function of mode interleaving was achieved in the time domain with a relatively small timing error, and the frequency of the pulse train could be increased to 40 GHz if a 10 GHz pulse train was injected. It should be noted that the the limited bandwidth and jitter of the oscilloscope may affect the accuracy of the timing measurement. For a more precise measurement, the set-up in [23] can be used. The power values of the four optical pulses in Figure 6b were different, mainly due to the following three reasons. First, the cascaded three optical combiners fabricated to simplify the testing process induced different losses for the four pulses. This can be eliminated by implementing an additional monolithic integrated PD for testing or combining a multi-mode on-chip modulator for practical optical ADC applications. Second, the overlaps between the four signals caused by the relatively wide input optical pulse would introduce power variations. This would cause interference when the abovementioned combiners combined the signals, which would result in variations in optical power. We also believe that this and the limited bandwidth of the link were responsible for the noticeable broadening in Pulse 1 and narrowing in Pulses 2, 3, and 4. Using a mode-locked laser could eliminate this impact. Third, the different propagation loss and mode conversion loss between the four pulses would also directly affect the optical power. The four pulses went through different lengths for the delay waveguide and different asymmetric directional couplers. Thus, the optical losses would be different. A non-uniform optical power splitter could be used to replace the 1 × 4 MMI coupler, which can compensate for the optical power differences.

## 4. Conclusions

In summary, we demonstrated an optical pulse interleaver utilizing time mode interleaving. The fabricated device converted an optical pulse into a 40 GHz pulse sequence composed of four pulses, successfully showing the interleaving performance. The RMS timing error was 0.9 ps. The device was composed of passive components, avoiding the use of active devices for wavelength tuning in the TWI schemes. Our proposed S-shaped delay path design reduced the total waveguide length by a factor of 46.3% compared with previously reported TWI schemes. The proposed TMI has the potential to be incorporated into optical ADC systems as a pulse multiplexing block for high-frequency sampling pulse generation.

## Figures and Tables

**Figure 1 micromachines-15-00326-f001:**
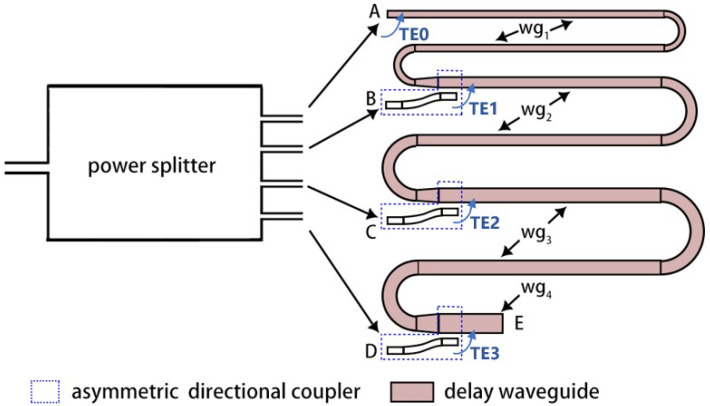
Schematic drawing of the TMI.

**Figure 2 micromachines-15-00326-f002:**
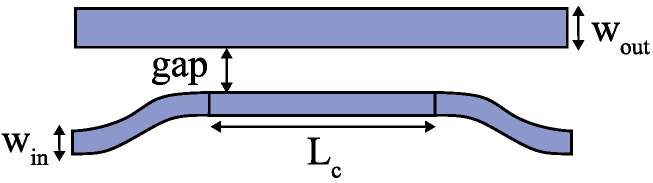
Schematic drawing of the asymmetric directional coupler.

**Figure 3 micromachines-15-00326-f003:**
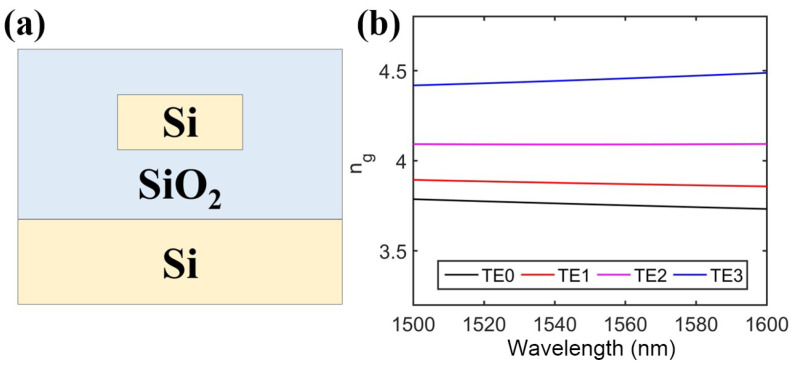
(**a**) Cross-sectional view of the strip waveguide. (**b**) Calculated group indexes of the four TE modes with a waveguide width of 1900 nm.

**Figure 4 micromachines-15-00326-f004:**
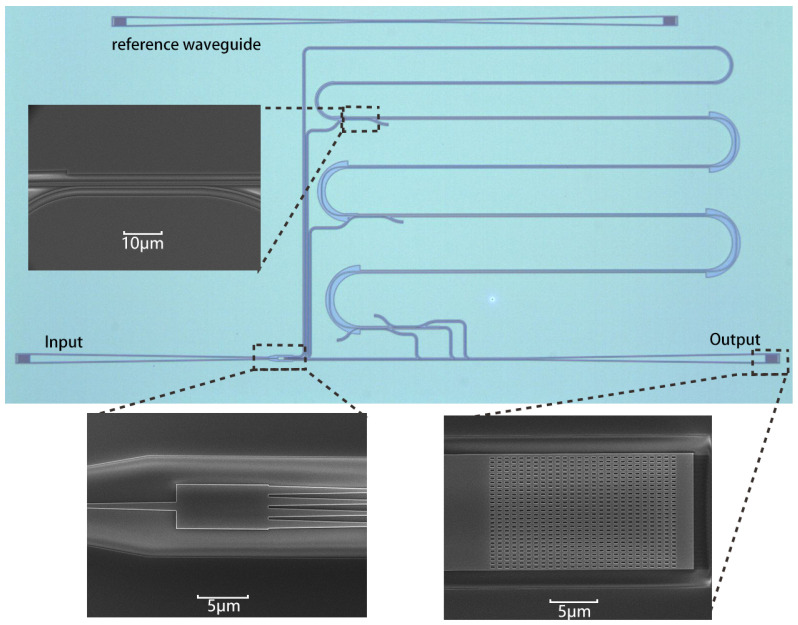
Microscope view and scanning electron microscope (SEM) view of the fabricated device.

**Figure 5 micromachines-15-00326-f005:**
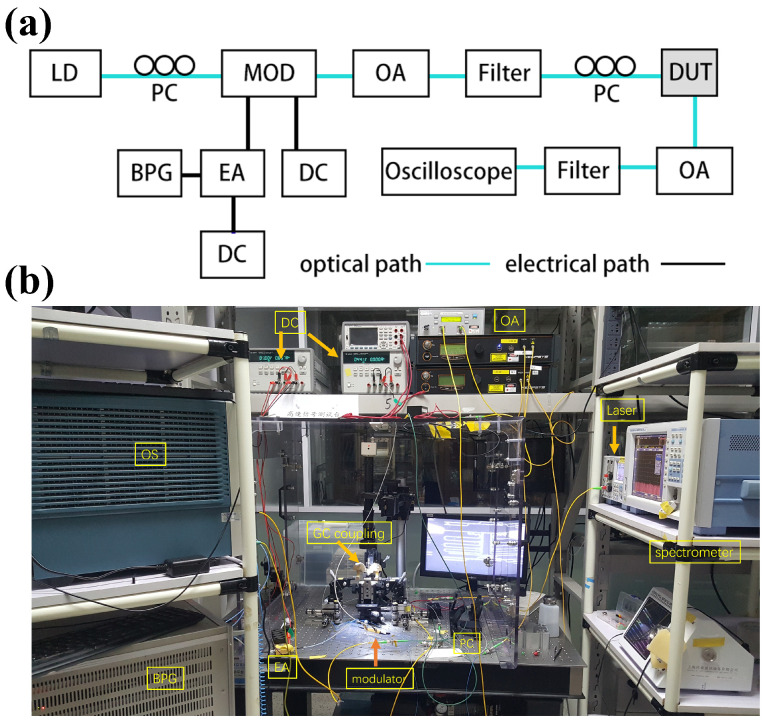
(**a**) Experimental set-up. (**b**) View of the testing environment. LD = laser diode; PC = polarization controller; MOD = modulator; OA = optical amplifier; DUT = device under test; BPG = bit pattern generator; EA = electrical amplifier; DC = direct current.

**Figure 6 micromachines-15-00326-f006:**
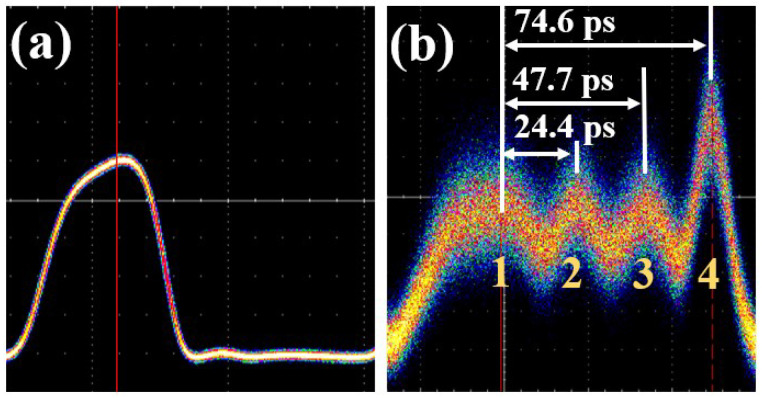
(**a**) Input pulse signal in the experiment. (**b**) Output pulse signal of the TMI.

**Table 1 micromachines-15-00326-t001:** Group indexes of optical modes in different waveguides.

	${\mathrm{wg}}_{450\mathrm{nm}}$	${\mathrm{wg}}_{930\mathrm{nm}}$	${\mathrm{wg}}_{1416\mathrm{nm}}$	${\mathrm{wg}}_{1900\mathrm{nm}}$
TE0	4.3027	3.8699	3.7863	3.7575
TE1	×	4.4005	4.0011	3.8735
TE2	×	×	4.4317	4.0895
TE3	×	×	×	4.4485

**Table 2 micromachines-15-00326-t002:** Relative time delay (ps) of different modes in waveguides.

	$w_{450\mathrm{nm}}$	$w_{930\mathrm{nm}}$	$w_{1416\mathrm{nm}}$	$w_{1900\mathrm{nm}}$	$T_{\mathrm{total}}$
TE0	29.5	24.1	21.3	0	75
TE1	×	27.4	22.6	0	50
TE2	×	×	25.0	0	25
TE3	×	×	×	0	0

## Data Availability

The data presented in this study are available on request from the corresponding author. The data are not publicly available due to ethical considerations regarding participant privacy during the study.

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
