# Peer review of "A Time Mode Pulse Interleaver in a Silicon-on-Insulator Platform for Optical Analog-to-Digital Converters"

_micromachines, 2024, doi:10.3390/mi15030326_

Round 1

Reviewer 1 Report

Comments and Suggestions for Authors

The manuscript entitled "Time-mode pulse interleaver in silicon-on-insulator for optical analog-to-digital converters" by Tu et al, reports an integrated time-mode pulse interleaver with a decreased length and low RMS timing error. The overall operation principle of the device is interesting and the results are promising. However, the authors need to address the points I raised here before the manuscript is considered to be published.

Some points to be addressed:

1.     In the Introduction section, the authors discuss various issues with optical ADCs extensively. However, the aspect more pertinent to this work is the generation of high-frequency optical signals. There seems to be insufficient coverage of previously reported work in this area. It is recommended that the authors supplement this section with additional information regarding these methods.

2.     In the section on TMI device design, the structure of the device is not clearly described. For instance, during the modal analysis and its results (as shown in Figure 1), what is the corresponding structure? Is there a cladding involved? This needs to be clarified for better understanding of the device configuration.

3.     On the Page 2, the authors state, "we can adjust the transmission time by controlling the propagation length." However, in reality, once the device is fabricated, its length becomes fixed. Therefore, it is not accurate to describe this structure as tunable. This point needs to be addressed or clarified in the manuscript.

4.     The design of the asymmetric directional coupler (ADC) is crucial to the device's performance, yet the manuscript does not present the design and optimization process of the ADC. This part should be supplemented. Additionally, how is the insertion loss of the ADC that has been fabricated? What is the conversion efficiency from the fundamental mode to higher-order modes? These aspects should also be addressed in the paper.

5.     An analysis of the modes of waveguides with different widths should be included to explain why the various widths mentioned in the paper were chosen for higher-order mode waveguides. This will help in understanding the rationale behind the selection of specific waveguide dimensions for the intended applications.

6.     The authors compare their work with Reference 15 and conclude that there is a 46.3% reduction in length. However, the signal repetition frequency and other conditions in Reference 15 are not the same as in the current paper. This raises doubts about the validity of such a comparison. I suggest the authors carefully reconsider the appropriateness of this comparison in their analysis.

7.     On the fourth page, the term "buried layer width" is used incorrectly. It should be replaced with "buried layer thickness".

8.     What is the overall insertion loss of the proposed device?

9.     What is the wavelength used in the experiments? Additionally, if the wavelength were to change, how would this affect the performance of the device? It is crucial to understand the impact of varying wavelengths on the device's functionality, as this could have significant implications for its practical applications.

10.  In Figure 6(a), the shape of the input waveform appears irregular, neither resembling a square wave nor a Gaussian profile. What is the reason for this irregularity?

11.  What is the y-axis in Figure 6? Also, why do the four optical pulses have different amplitudes, and why is there a noticeable broadening in Pulse 1? These details are crucial for a comprehensive understanding of the results presented in Figure 6 and their implications for the performance of the device under study.

Reviewer 2 Report

Comments and Suggestions for Authors

An interesting work both in design and fabrication. However, I figure the results should be revised, as the PD and the oscilloscope can not provide response time down to less than 0.1ps, which are usually only several ps to tens ps. For high precise time scaling, please read the following paper:

"Highly precise timing alignment of multi-wavelength interleaved cavity-less pulse sources with FROG." Opt. Express 31 (26): 44515--44522.

Reviewer 3 Report

Comments and Suggestions for Authors

The paper entitled “Time-mode pulse interleaver in silicon-on-insulator for optical analog-to-digital converters” presented a fabricated optical pulse interleaver based on time-mode interleaving for optical ADC system applications. Overall, this paper shows the novel design and results of the devices, thus I would like to recommend it to be published in Micromachines after address the following concerns:

(1)   In terms of the structure of this paper, I would suggest the authors to adjust the structure of Section 2. Design of the TMI, for example, change the order of Figure 1 and Figure 2&3 and their relevant contents. I believe this arraignment will benefit for potential readers for a more clear clue of the motivation and consistency.

(2)   Could the authors explain more about why they choose the specific four wavelength or waveguide modes for the TMI design in page 4?

Comments on the Quality of English Language

Some minor English issues needed corrected.

Round 2

Reviewer 1 Report

Comments and Suggestions for Authors

The name of the label of the x-axis in Figure 3(a) should be "Wavelength (nm)".